# The PAIR-R24M Dataset for Multi-animal 3D Pose Estimation

**Jesse D. Marshall**\*
Organismic and Evolutionary Biology
Harvard University
Cambridge, MA 02138
jesse_d_marshall@fas.harvard.edu

**Ugne Klibaite**\*
Organismic and Evolutionary Biology
Harvard University
Cambridge, MA 02138
klibaite@fas.harvard.edu

**Amanda Gellis**
Organismic and Evolutionary Biology
Harvard University
Cambridge, MA 02138
agellis@fas.harvard.edu

**Diego E. Aldarondo**
Organismic and Evolutionary Biology
Harvard University
Cambridge, MA 02138
diegoaldarondo@g.harvard.edu

**Bence P. Ölveczky**
Organismic and Evolutionary Biology
Harvard University
Cambridge, MA 02138
olveczky@fas.harvard.edu

**Timothy W. Dunn**
Department of Biomedical Engineering
Duke University
Durham, NC 27708
timothy.dunn@duke.edu

## Abstract

Understanding the biological basis of social and collective behaviors in animals is a key goal of the life sciences, and may yield important insights for engineering intelligent multi-agent systems. A critical step in interrogating the mechanisms underlying social behaviors is a precise readout of the 3D pose of interacting animals. While approaches for multi-animal pose estimation are beginning to emerge, they remain challenging to compare due to the lack of standardized training and benchmark datasets. Here we introduce the PAIR-R24M (Paired Acquisition of Interacting oRganisms - Rat) dataset for multi-animal 3D pose estimation, which contains 24.3 million frames of RGB video and 3D ground-truth motion capture of dyadic interactions in laboratory rats. PAIR-R24M contains data from 18 distinct pairs of rats and 24 different viewpoints. We annotated the data with 11 behavioral labels and 3 interaction categories to facilitate benchmarking in rare but challenging behaviors. To establish a baseline for markerless multi-animal 3D pose estimation, we developed a multi-animal extension of DANNCE, a recently published network for 3D pose estimation in freely behaving laboratory animals. As the first large multi-animal 3D pose estimation dataset, PAIR-R24M will help advance 3D animal tracking approaches and aid in elucidating the neural basis of social behaviors.

---

\*Equal contribution.

35th Conference on Neural Information Processing Systems (NeurIPS 2021) Track on Datasets and Benchmarks.

# 1   Introduction

Social behaviors are core components of an animal's behavioral repertoire. Understanding their neural, biological, and evolutionary basis has long been a focus of the life sciences [1, 2] and may inform treatments for psychiatric diseases, such as autism spectrum disorder and schizophrenia, where social interactions are impaired [3, 4].

Precisely phenotyping social behaviors and identifying their neural basis requires reliable and quantitative measures of social behavior in animal models [5]. Currently, studies largely rely on scoring performance in highly structured assays, for instance the tube test, 3 chamber test, or resident-intruder test [6]. While these provide interpretable readouts, they are ethologically limited and compress complex behavioral processes into scalar variables of questionable biological significance [7]. In contrast, assays in unrestrained animals that use computer vision and behavioral classification offer the ability to profile a richer range of social behaviors between animals, but are more challenging to quantify and interpret [8–12].

To improve behavioral quantification, convolutional neural networks for automated detection of an animal's 2D pose [13–15], and more recently 3D pose [16–18], have been developed. However, in comparison to single animal tracking, methods for multi-animal postural tracking, especially in 3D, are only beginning to emerge. Existing 2D pose recognition techniques employ a mixture of 'top-down' multi-animal tracking, in which pose is reconstructed within identified bounding boxes of multiple animals (e.g [8, 14]) and 'bottom up' architectures that first detect all body landmarks and then assign them to animals [19–21]. Both top-down and bottom-up multi-animal tracking approaches are promising, but need substantial amounts of training data to accurately track animal pose in the face of challenging occlusions generated by socially interacting animals.

Development of new data-efficient and occlusion-robust multi-animal tracking approaches requires standardized pose estimation datasets and benchmarks, which do not exist in 3D. To address this, we introduce PAIR-R24M, a novel dataset relating multi-view color video and ground-truth 3D kinematics in behaving rats. We collected over 24 million frames of 30 Hz color video across 24 camera views in 18 different pairs of rats interacting in a behavioral arena. In each frame, a motion capture system provides the 3D positions of 12 body landmarks on each individually identified animal, describing the movement of its head, trunk, shoulders, and hips. Each frame is associated with a behavioral label, denoting which of 11 behavioral categories and 3 inter-animal interaction categories it matches best. These labels can be used to balance datasets during training, rigorously assess pose estimation performance over a wide variety of poses, provide labels for action recognition approaches, and perform detailed analyses of behavioral patterns.

# 2   Related Work

## 2.1   Datasets for single and multi-animal 3D pose

There exists a small collection of publicly available 3D animal pose benchmark datasets. The Acino dataset contains 7,588 frames of hand-labeled 3D poses (20 keypoints) from cheetahs, capturing mostly running behaviors [22]. The Open Monkey Studio dataset contains 195,228 hand-labeled frames (13 keypoints) of macaques in a large, enriched enclosure across 62 camera views [16]. Two other approaches use motion capture systems to provide expanded 3D ground-truth datasets. RGBD-Dog includes 3D keypoint data (63-82 keypoints from motion capture) and depth maps along with 8-10 RGB video views in canines, although is limited to 5 behaviors [23]. Rat 7M contains nearly 7 million frames and 3D keypoints across a wide range of rat poses, providing a powerful substrate for training and testing algorithms in rodents, the most common model organisms in biomedicine [18]. While valuable, each of the datasets is limited to individual animals.

Thus far, multi-animal datasets exist only for 2D. Graving et al. released videos and 2D annotations for large groups of locusts and zebras filmed from a single top-down view [14], providing valuable datasets for benchmarking 2D collective behavior tracking algorithms. Pereira et al. published a set of labeled fruit fly courtship data [20]. Lauer et al. released annotated multi-animal datasets from mice, mouse pups, marmoset, and zebrafish [21]. By far the most extensive multi-animal 2D dataset is CalMS21, which was released as part of the Multi-Agent Behavior Challenge 2021 and consists of 6 million frames of unlabeled and over 1 million frames of tracked poses and behavioral annotations of pairs of interacting mice [24].

In the more mature field of 3D human pose estimation, many multi-human 3D datasets are available, which vary broadly in the number of behaviors tracked, number of cameras used, means of marker tracking, and environmental context. The CMU Panoptic dataset provides 480 camera views during a wide range of social behaviors in a laboratory environment, with 3D poses obtained via pose estimation [25]. The Campus, Shelf (manually annotated) and MuPoTS-3D (derived from pose estimation) datasets offer 3D poses and multi-view video in real-world scenes [26,27], while 3DPW offers monocular footage with 3D pose labels derived from inertial measurement units [28]. The MuCo-3DHP dataset [27] is a large, multi-human 3D dataset generated by splicing together individual subjects, and their ground-truth markerless annotations, from the expansive MPI-INF-3DHP dataset [29]. Other benchmark datasets exist in specific domains, such as stores [30] and operating rooms [31]. Others use synthetically rendered humans [32–36] or body surfaces [37]. Together these datasets have fueled a productive era of 3D pose tracking, but their domain is drastically different from laboratory animals. Developing the type of 3D animal tracking algorithms required to accelerate progress in neuroscience, biomedicine, and ecology will require in-domain datasets that permit relevant training and benchmarking over a diversity of body plans and behaviors.

## 2.2 Algorithms and benchmarks for animal 3D Pose Estimation

To our knowledge there is only one example of multi-animal 3D pose estimation in the literature [16], likely due to the lack of large training and benchmark datasets in this domain. There are several existing algorithms for 3D pose in individual animals. DANNCE [18] and Freipose [17] use volumetric representations of multi-view inputs to combine image features across cameras and enable 3D supervision, similar to the current state-of-the-art for multi-view human pose [38]. 3D DeepLabCut [22,39] uses triangulation of 2D detections across multiple views, which GIMBAL [40] and Anipose [41] further refine using spatiotemporal constraints. Open Monkey Studio uses a triangulation-based method but with a larger set of cameras, and in addition to using spatiotemporal constraints, makes use of reprojections into unlabeled views to increased their labeled training pool [16]. DeepFly3D uses triangulation, bundle adjustment, and pictorial structures to provide robust 3D pose estimation in tethered flies [42]. For monocular 3D pose estimation, "lifting" approaches using a fully connected network to infer 3D pose from 2D estimates [43,44] have been extended from work in humans [45] to tethered flies and lab mammals. In addition to lifting, Bolaños et al. [43] use synthetic data to improve 3D pose detection in restrained mice. As of yet, none of these methods have been extended to multi-animal 3D pose estimation. In this study we extend the DANNCE volumetric approach because it has demonstrated superior performance on rodents compared to multi-view triangulation, and also because multi-view triangulation would be further complicated by errors in multi-animal identity tracking.

## 2.3 Multi-animal action recognition

We follow the lead of human 3D pose datasets and group our data into standardized behavioral categories to aid the training and benchmarking of pose-estimation and action-recognition algorithms. However, unlike traditional 3D pose datasets acquired using human actors given explicit instructions, here we needed to infer behavioral categories from movement by extending 3D action recognition methods to the multi-animal setting. Multi-animal action recognition has remained challenging due to a lack of ground-truth and, relatedly, a lack of intuition about the definitions and structure of animal behavior, especially in social contexts. Existing methods for multi-animal action recognition employ supervised learning using human-labeled behavior categories such as mounting or attacking, classified using a variety of features describing behavior: pixels [46], the set of 2D body landmarks visible from a single top-down views [8,24], hand-designed features of 2D body landmarks [47] (sometimes supplemented with depth imaging information [48]), shapes fit to 2D or 3D body contours [9,49], or quantities derived from movement trajectories, such as velocity and heading direction [50]. Other methods use unsupervised learning techniques, again on a range of behavioral features: pixels [51], 2D pose features [12] or both [30]. While no approach has performed unsupervised analysis of multiple animals using 3D pose, Marshall et al. [52] designed an approach for identifying behaviors in single animals based on 3D pose features. Here we extend this approach to multiple animals, and create inter-individual features to define new interaction behavioral categories. This straightforward, yet effective, unsupervised action recognition approach allows us to segment and balance the PAIR-R24M dataset and introduce a foundational algorithm applicable to new multi-subject 3D pose data across species.

# 3    Dataset and Benchmark

## 3.1    The PAIR-R24M dataset

To collect the PAIR-R24M dataset we used CAPTURE, a technique that uses body piercing to chronically attach retro-reflective markers to small animals, allowing their pose to be reconstructed using motion capture [52]. We attached 12 markers to the dorsal surface of each animal at identical locations to label their head, trunk, hips, and shoulders. We additionally added 1-2 markers to the head and trunk of animals to differentiate individuals. If interacting animals bore identical marker sets, we masked a marker on the head using whiteout to disambiguate them.

We used a 12 camera motion capture array to record the position of the markers at 300 Hz with sub-mm precision (Fig. 1A). We used commercial Cortex (Motion Analysis) software, which utilizes pairwise distances between markers and a parametric body model, to assign marker identities to each animal. We concurrently recorded animals at 30 Hz using 6 RGB video cameras. We calibrated the video cameras into the same world coordinate system as the motion capture array to automatically label video frames by projecting the 3D marker positions.

We then performed simultaneous CAPTURE and video recordings for 18 pairs of animals ($n$=7 subjects bearing markers, $n$=2 markerless subjects), for 1 hour each (108,000 timepoints). To increase viewpoint diversity, we moved each of the video cameras to 4 different locations across recordings (Fig. 1B). On a subset of camera views and frames in which animals were rapidly moving, we noted discrepancies between motion capture and video due to slight errors in synchronization and calibration (Appendix 3). While these errors could in the long term pose limits in the precision of the dataset as a benchmark, they occur on a limited subset of frames, and similar discrepancies exist in commonly used human 3D pose datasets [38].

We recorded from a subset of animal pairs in each recording condition, yielding a total of 26 hours of data of paired animals bearing markers. We also recorded 14 hours of data from animals bearing markers when paired with animals not bearing markers, to add additional markerless video frames to the dataset. These single-markerset recordings also allowed us to assess the fidelity of animal identity assignment in the dataset. Head segment lengths, which were constant within subjects but differed slightly between subjects due to small changes in head marker placement during headcap construction, were stable across individual animals when compared over single- and double-markerset paired recordings (Appendix 4). Additionally, we recorded from each subject alone for 30 minutes to facilitate the construction of single animal tracking models, and recorded from individual and paired animals not bearing markers. Single animal and paired markerless video recordings are not included in the present dataset but may be added at a later data to facilitate transfer and benchmarking of semi-supervised tracking approaches.

Occasionally, self-, animal-animal, or environmental occlusions prevented 3D marker tracking by the motion capture system. As most of these periods were temporally succinct, we imputed missing data using linear interpolation within an egocentrically aligned reference frame anchored on the animal's center of mass and rotated to place the front of the animal's spine along the y-axis. The center of mass and orientation of the animals were estimated from the remaining markers if spine markers were absent. We also sometimes observed other errors where the motion capture system incorrectly assigned marker position. We addressed incorrect assignment by flagging potential errors using a $4\sigma$ threshold on z-scores of inter-marker distance, although we note these frames still appeared to possess accurate behavioral categorization. Our official 24M dataset size is calculated after excluding any frame with at least one flagged marker. In the released dataset, we provide all recorded frames, together with z-scores for each marker, permitting researchers to use partially tracked frames if desired.

## 3.2    Action recognition

The performance of human and animal pose tracking algorithms can vary widely depending on the behaviors animals perform — for instance highly-occlusive rodent grooming behaviors are often challenging to reconstruct — making it important to assess the performance of tracking algorithms in an action-specific manner. There remains no standard taxonomy of rodent behaviors [53], and there is often disagreement among human observers about what defines a behavior and when they begin and end (e.g [24, 54]). We therefore used an unsupervised approach to identify behaviors by

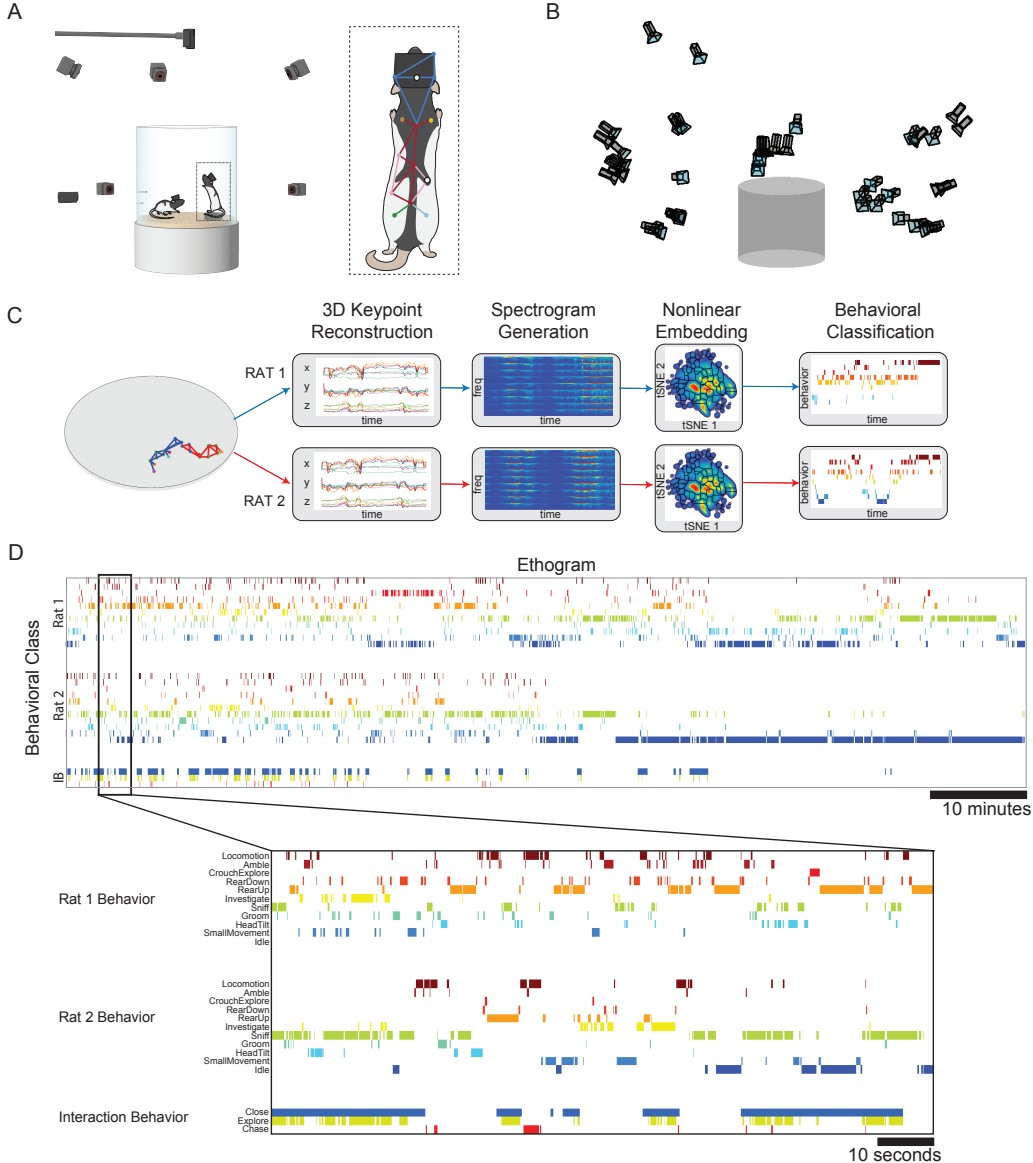

Figure 1: (A) Schematic of the recording arena with two interacting rats surrounded by motion capture and video cameras. Inset: location of the recording markers used on the animal's body. White markers indicate markers that were tracked in a subset of animals to distinguish between animal pairs, but are not used for further analysis. (B) Location of the video camera positions relative to the recording arena across all experiments. Each camera was moved to 4 different locations during acquisition of the dataset, but there were additional minor shifts in position across recordings (42 total positions across all cameras). (C) Analysis pipeline schematic for applying behavioral labels given 12-point skeletons obtained from motion capture recordings. (D) Ethograms for individual and interaction behavioral categories for a sample 1-hour movie. Expanded region corresponds to a two-minute behavioral bout. Example Movies.

|  | Subj. 1 | Subj. 2 | Subj. 3 | Subj. 4 | Subj. 5 | Subj. 9 | Subj. 10 | All |
|---|---|---|---|---|---|---|---|---|
| Pairs | 6 | 6 | 6 | 6 | 6 | 2 | 2 | 18 |
| $CL_{50}$ | 156 | 110 | 90 | 134 | 240 | 178 | 128 | 138 |
| $CL_{95}$ | 2642 | 1200 | 1300 | 1281 | 2433 | 1676 | 1283 | 2593 |
| IB1 (Close) | 1.42M | 1.19M | 1.32M | 1.23M | 872k | 307k | 307k | 3.32M |
| IB2 (Explore) | 797k | 763k | 823k | 752k | 448k | 164k | 164k | 1.96M |
| IB3 (Chase) | 29.6k | 32.4k | 33.4k | 42.1k | 24.2k | 21.8k | 21.8k | 103k |
| B1 (Idle) | 2.69M | 1.95M | 1.84M | 1.80M | 2.52M | 377k | 1.01M | 12.2M |
| B2 (SmallMovement) | 897k | 707k | 484k | 488k | 394k | 224k | 105k | 3.30M |
| B3 (HeadTilt) | 399k | 260k | 268k | 223k | 428k | 166k | 183k | 1.93M |
| B4 (Groom) | 458k | 606k | 319k | 302k | 290k | 210k | 120k | 2.31M |
| B5 (Sniff) | 1.32M | 1.71M | 1.01M | 1.25M | 1.50M | 421k | 372k | 7.59M |
| B6 (Investigate) | 535k | 438k | 246k | 319k | 155k | 169k | 60.4k | 1.92M |
| B7 (RearUp) | 896k | 736k | 787k | 638k | 253k | 236k | 119k | 3.66M |
| B8 (RearDown) | 223k | 215k | 200k | 228k | 126k | 153k | 90.6k | 1.24M |
| B9 (CrouchExplore) | 230k | 67.5k | 206k | 122k | 34.6k | 65.5k | 28.4k | 755k |
| B10 (Amble) | 101k | 101k | 90.5k | 109k | 111k | 61.8k | 54.4k | 628k |
| B11 (Locomotion) | 235k | 214k | 202k | 222k | 180k | 288k | 138k | 1.48M |
| Total Frames | 7.98M | 7.00M | 5.65M | 5.71M | 5.99M | 2.37M | 2.28M | 24.3M |

Table 1: Recording summary statistics for all animal subjects. Pairs is the total number of unique animal pairs recorded for each subject. ($CL_{50}$) 50th percentile of contiguous clip length (in frames) after excluding frames with at least one poorly tracked marker in both animals; ($CL_{95}$) 95th percentile of contiguous clip length (in frames); (IBx) frames for interaction behaviors; (Bx) frames for individual behaviors. The "All" column tallies over unique items (e.g. Subj. 2 + Subj. 1 IB only counted once).

first clustering pose dynamics in a reduced dimensional behavioral feature space, and then manually inspecting samples from each cluster to assign cluster names *post hoc*, following previously published approaches [52, 55]. To cluster the animals' behavior, we first performed principal component analysis on the all-to-all marker distances across all frames. We applied a Morlet wavelet transform to the top 10 principal components at 25 dyadically spaced frequencies from 0.5-20 Hz. These features, along with the z-heights and local smoothed velocities of each marker, composed a feature vector. To balance the clustering, we applied tSNE separately to each recording and sampled 1,000 frames distributed evenly across the behavioral embedding of each reduced dataset [12]. We then concatenated the sampled frames from each dataset and embedded them with tSNE, resulting in a comprehensive, balanced embedding space of all animal behavior in the dataset. We then re-embedded wavelet values from each movie using convex optimization, as described in [55], transformed the map into a density distribution after smoothing it with a Gaussian kernel, and applied a watershed transform to divide the data into discrete clusters.

The number of behavioral clusters identified in the embedding space can be varied by changing the density kernel used to create the space. We provide two resolutions of behavioral labeling in the dataset. First, a set of 11 coarse behavioral categories that can be used to balance the dataset and benchmark algorithms across different behaviors. Second, a set of 84 fine behavioral categories that can be used for a more detailed analysis of the animal's behavior.

The coarse behavioral categories reflected common classes of rodent behavior, including rearing, locomotion, and investigation (Fig. 2), each of which results from a manual clustering of fine-grained clusters. Within these fine behavioral categories across the full dataset, behaviors varied in frequency by several orders of magnitude, from ~6,000 to ~6,000,000 time-points (Fine Behavior 62 – a side-to-side head sweep vs. Fine Behavior 35 – a high-frequency sniff). This class imbalance highlights the importance of obtaining large datasets to train and benchmark behavioral tracking algorithms, especially if algorithm performance on rare behaviors is desired.

To further isolate different classes of inter-animal interactions, we further divided periods in which animals' centroids were within one body length of one another (200 mm) into three different interaction behavioral categories: synchronized locomotion ("Chase"), stationary exploration ("Explore"; when both animals were in any coarse behavioral category among HeadTilt, Groom, Sniff, Investigate, Rears, and CrouchExplore), or other times when animals were adjacent ("Close"). Because

inter-animal interactions contain numerous occlusions, they represent a challenging use case for multi-animal tracking algorithms. The over 5.3 million frames of animal interactions we provide here provide an ample diversity of frames to train and benchmark new pose tracking algorithms in social settings.

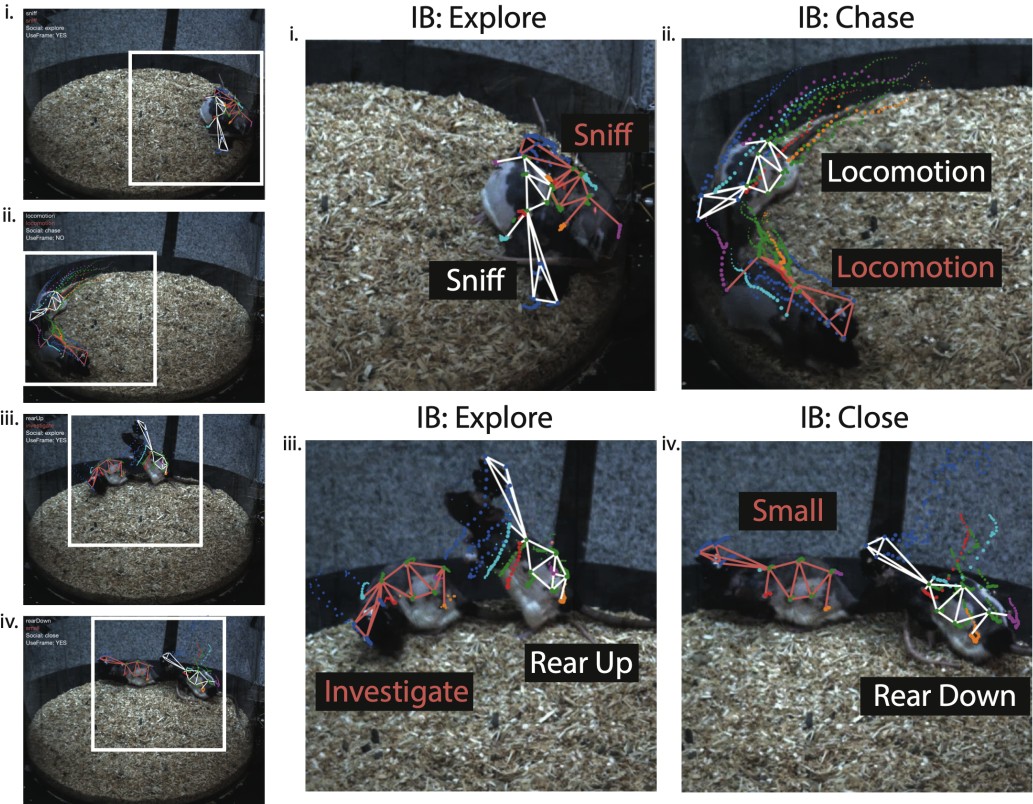

Figure 2: Example reprojections of ground-truth motion capture onto single camera views, shown for specific behavioral categories (pink and white labels corresponding to each rat skeleton) and interaction behavior categories (IB). Trailing points illustrate past 1-second trajectories for each marker. Example Movies.

The video frames, 3D pose estimates, and behavioral annotations are continuous in time, with only moderate interruptions in pose estimates due to flagged tracking errors. The median length of continuously tracked snippets is 138 frames ( 4.5 s), with a long tail such that 5% of all continuous snippets are greater than 86 s in length (Table 1). This will be useful both for benchmarking video-based pose tracking algorithms that use local temporal information [56], as well as building statistical models of single and multi-animal behavior [57, 58]. As an example of their use for analyzing the mathematical structure of behavior, we can visualize the ethograms of each animal's behavior, which show that animals transition over many individual and interacting behaviors during a recording session (Fig. 1D).

## 3.3 DANNCE benchmark

To establish baseline benchmarks for pose estimation to which future algorithms should be compared, we used a multi-animal extension of DANNCE [18], the current state-of-the-art for rat 3D pose estimation. Because DANNCE's standard mechanism is to encapsulate a subject in a 3D volumetric bounding box via geometric sampling of multi-view image content, multi-animal inference was performed simply by running each animal's 3D volume through the network independently (see Appendix 5 for details). When animals are separated in space, such that their 3D volumes are

non-overlapping, this approach trivially reduces to the single animal case. When animals are nearby and overlapping, however, DANNCE must overcome significant animal-animal occlusion and infer correct landmark-subject associations. Our dataset provides a large library of interacting behavior examples that DANNCE, and other approaches, can use to learn social-specific poses and complex, multi-animal image features.

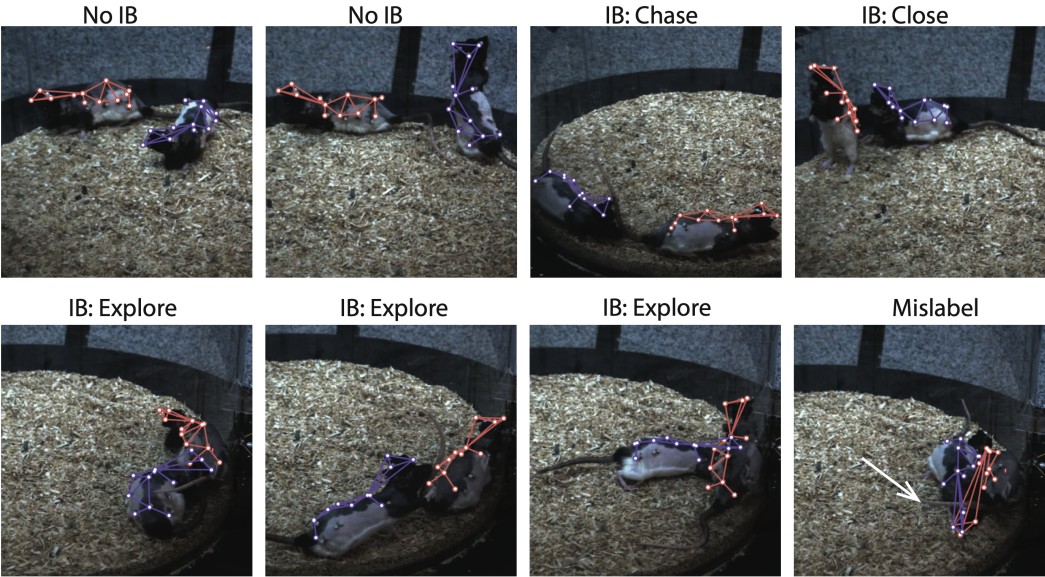

Figure 3: Example DANNCE predictions, reprojected onto a single camera view, for specific interaction behavioral categories. Example predictions are on validation recordings, with the validation animal (Subject 5) in red. Each frame is labeled with the interaction behavior category. The white arrow in the "Mislabel" panel points to an error in head identity prediction. Example Movies.

|            | $MPJPE_H$ | $MPJPE_T$ | MPJPE | $PJPE_{50}$ | PCK@0.5 | PCK@0.75 | mPCK |
|------------|-----------|-----------|-------|-------------|---------|----------|------|
| DANNCE.L2* | 6.44      | 9.02      | 8.37  | 7.63        | 0.68    | 0.89     | 0.88 |
| DANNCE.L2  | 4.87      | 7.93      | 7.17  | 6.44        | 0.79    | 0.94     | 0.93 |
| DANNCE.L1* | 4.28      | **7.13**  | **6.41** | **5.77**  | 0.83    | 0.96     | 0.95 |
| DANNCE.L1  | **4.22**  | 7.35      | 6.57  | 5.87        | 0.83    | 0.96     | 0.95 |

Table 2: DANNCE 3D multi-animal pose estimation benchmarks. In DANNCE.$X$, $X$ indicates the type of loss function used for training. (*) was trained from a random initialization of weights, and the others from a network pre-trained on Rat 7M [18]. ($PJPE_{50}$) 50th percentile of the per joint prediction error (in mm), i.e. the Euclidean distance between predicted and ground-truth markers. (MPJPE) mean PJPE, also broken down by head ($MPJPE_H$) and trunk ($MPJPE_T$). (PCK@0.5) percent correct keypoints using a distance threshold of 50% of the distance between two head markers. (PCK@0.75) PCK using a threshold of 75% of the distance. (mPCK) mean PCK over 11 equally spaced thresholds.

We trained DANNCE for 30 epochs, using 420k images (70k poses) per epoch, and varied the pretraining conditions and type of loss function to measure the influence of these parameters on performance. Our results on withheld validation subject 5 are presented in Table 2. When using DANNCE's previously published L2 loss function, DANNCE performance improved with pretraining on Rat 7M. However, training with an L1 loss, with or without pretraining, ultimately minimized the mean per joint prediction error (MPJPE) across all markers (additionally broken down by head, $MPJPE_H$, and trunk, $MPJPE_T$) and maximized percent correct keypoints (PCK) at all distance thresholds (@ fractions of the distance between two head markers – 19.4 mm). Across behaviors, DANNCE tracked Investigate with the smallest and CrouchExplore with the the largest error, respectively, although error was within 10% across most behavioral categories (Appendix Table 3). DANNCE performed similarly well on all close social interaction behaviors (Appendix Table 4). Qualitatively,

DANNCE generally made remarkably consistent landmark predictions even in periods of spatial overlap between animals, but it did sometimes briefly assign head landmarks to the wrong animal during specific close interaction poses (Fig. 3).

## 4   Limitations

Our dataset will already be a valuable resource for social behavioral tracking, but there are several present limitations that could be addressed in future work. First, due to frequent occlusions in the multi-animal settings, there are periods without accurate landmark tracking that we dropped from the dataset. Future datasets could incorporate a larger number of cameras to reduce the number of missing data periods. Second, the ground-truth motion capture data comes from a reduced 12-marker set that does not capture points on the distal limbs, and this could contribute to a loss of precision in behavioral identification. One potential solution for limb tracking is to train using a combination of the 20-marker Rat7M, which includes multiple limb markers, and PAIR-R24M datasets. Limb keypoints could also be added to the dataset using a combination of manual labeling, e.g. through crowdsourced annotation, and inference, similar to datasets like CMU Panoptic [25]. However, annotating keypoints in animals is generally more challenging for non-primate species, where identification of body parts requires more domain knowledge, making the use of crowd-sourced annotation platforms challenging.

## 5   Discussion

The PAIR-R24M dataset is the largest and most diverse benchmark dataset for the rapidly growing field of multi-animal behavioral measurement and analysis. We make the dataset available for researchers interested in training new multi-animal tracking and action recognition algorithms, and for researchers interested in mining the data for new quantitative insights on the nature of social behavior. Specifically, we expect that this dataset will help to address the problems of multi-animal 3D pose estimation and instance segmentation.

In our dataset we solve instance segmentation by identifying individuals using known differences in their respective marker sets. These ground-truth animal identities will assist in the development and evaluation of deep learning algorithms that identify individuals through either top-down inference, such as convolutional networks for identity detection or center-of-mass tracking (e.g. [20, 59, 60]), or bottom-up inference such as 3D extensions of part affinity fields [61].

The PAIR-R24M dataset should also help develop new approaches for multi-animal 3D pose estimation. Here, we performed pose estimation using a state-of-the-art volumetric animal pose tracking approach. While our approach was generally effective, it made mistakes on some types of close interaction, a relevant concern considering that most interesting social behaviors are characterized by profound animal-animal overlap and contorted poses. Our results may be improved by newer architectures that employ semi-supervised learning or temporal convolutions [56] in addition to previously discussed bottom-up methods. Additionally, while highly performant, the use of volumetric convolution is computationally expensive, limiting inference speeds. PAIR-R24M will aid the development and evaluation of new, fast and performant multi-view 3D pose estimation algorithms.

While the PAIR-R24M dataset is an important step in the collection and dissemination of benchmarks for animal pose estimation, it can be extended in many ways. While we used motion capture as a high-throughput means of collecting training data, labels for animal hands, feet, and other appendages will be necessary for training algorithms that predict more complete descriptions of animal movement. These labels could come from human annotators [18], and crowd-sourcing efforts have begun to assemble such detailed annotations for animals in 2D (e.g. [62]; although see Section 4). Datasets extending beyond keypoints to capture an animal's full 3D body surface, as is now possible in human subjects, will also be valuable. While 3D scans have been used to assemble parametric body models of animals in specific poses [63], the databases that are available are still small compared to those available in humans [36, 64] and do not contain data from freely moving subjects. While cross-domain adaptation approaches [43, 62, 65] may facilitate some progress in 3D surface estimation, ground-truth databases are needed to appropriately benchmark and train these techniques. Finally, future datasets from other species, environments, and social contexts will help to build algorithms that are flexible across a rich array of tasks and contexts, with the ultimate goal of enabling methodologies for full reconstruction of animal kinematics in complex, occlusive environments, with as few as one camera.

## Acknowledgements

J.D.M. acknowledges support from the Helen Hay Whitney Foundation and NINDS (K99NS112597), D.E.A. from NSF (DGE1745303), B.P.Ö from SFARI (646706), NIH (R01GM136972), and the Starr Family Foundation., and T.W.D. from NIH (R01GM136972) and the McKnight Foundation Technological Innovations in Neuroscience Award.

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
