# OpenReview forum: "The PAIR-R24M Dataset for Multi-animal 3D Pose Estimation"
_NeurIPS.cc/2021/Track/Datasets_and_Benchmarks/Round1 — NeurIPS 2021 Datasets and Benchmarks Track (Round 1)_

### Official Review · Reviewer_Q6LM · 2021-07-01

**Rating:** 6
**Confidence:** 3
**Correctness:** All the claims made in the submission…
**Clarity:** The paper is well written.

**Strengths:**

+ The paper is well written and is easy to read.
+ The paper provides large scales 3D pose annotations for rats.
+ The paper also provides behaviour annotations.
+ The way to collect annotations is clearly elaborated.
+ The statistics details of the dataset is provided.

**Weaknesses:**

- The authors mention that they recorded extra data when paired with animals not bearing markers, which can prevent swaps of motion capture markers across animals. (Line 156-158) Does it mean that currently, there exist lots of swap problems in the dataset? I think it will damage the reliability of this benchmark.
- Is it necessary to provide behavioural labels in each frame? I think not all motion in a time period relates to the labelled behaviour.
- The occlusions that lasted for over 3s will be dropped. The data preserved in the final datasets will be the cases that are less occluded. Therefore, the benchmark will be not challenging enough.

**Additional Feedback:**

Line 261, the authors mention human annotations of distal limbs can be added in future datasets. But I think manually annotating the 3D keypoints is quite difficult, especially under sub-mm accuracy. Is there any technique to ease this problem?

**Documentation:**

The documentations should be further updated.

**Ethics:**

I do not see any ethical concerns in this paper.

**Relation To Prior Work:**

It is clearly discussed how this work differs from previous contributions.

**Summary And Contributions:**

This paper proposes a new benchmark for multi-animal 3D pose and behaviour estimation. It is the first benchmark to study multi-animal in 3D cases, while providing individual and interaction behavioral labels.

---

> ### Author Response · Authors · 2021-07-13
> **Author Response**
>
>
> We appreciate the reviewer’s careful reading of the manuscript. We address the concerns below in full, and we will incorporate these responses and the proposed experiments on distal limbs in the manuscript.
>
> > *The authors mention that they recorded extra data when paired with animals not bearing markers, which can prevent swaps of motion capture markers across animals. (Line 156-158) Does it mean that currently, there exist lots of swap problems in the dataset? I think it will damage the reliability of this benchmark.*
>
> ### Swap quantification:
> While we noted swaps as a principal reason for including the marker-markerless data, we now believe this is a negligible source of error in the dataset. In a recent analysis of the distance between markers, we found that close animal interaction was not associated with  changes in the distribution of inter-marker distributions. This suggests that marker swaps are minimized. We will include these results and also describe additional utility of our  marker-markerless data: they expand the overall size of the dataset and potentially  expand the diversity of interaction behaviors included in the dataset
>
> > *Is it necessary to provide behavioural labels in each frame? I think not all motion in a time period relates to the labelled behaviour.*
>
> ### Behavioral categorization:
> In the original manuscript, we could have been clearer regarding the choice of the number of behaviors used (see response to Reviewer 1). Rather than representing extremely stereotyped behaviors, as in an action recognition dataset, the behavioral labels refer more broadly to categories of behavior, such as ‘Slow Locomotion’, ‘Fast Locomotion’, and ‘Rearing. Because these are broad and contain some diversity of different behaviors, every time point can be assigned to one of the categories.
>
> > *The occlusions that lasted for over 3s will be dropped. The data preserved in the final datasets will be the cases that are less occluded. Therefore, the benchmark will be not challenging enough.*
>
> ### Occlusions:
> We agree that the extended occlusions that are removed may remove some examples of animal-animal interactions from the dataset. However we still believe the dataset represents a considerable challenge as a benchmark:
> 1. Many examples of interacting animals are successfully tracked (e.g. Figure 2), suggesting that the current dataset is sufficient for learning challenging poses.
> 2. The keypoints are tracked using 12 very sensitive motion capture cameras, in which a marker needs only be seen in two cameras to be successfully triangulated. In contrast, we only use 6 video cameras to record animals. Thus there are many keypoints tracked using motion capture that are occluded in the RGB video cameras, and their reconstruction will be especially challenging using 3 or 1 views.
> 3. We see an improvement in DANNCE performance between using 64 voxels and 80 voxels, indicating that performance on this benchmark is still likely limited by the algorithms we have developed for pose tracking.
> Please also see our “Inter-animal occlusion” response to Reviewer 1.
>
> > *Line 261, the authors mention human annotations of distal limbs can be added in future datasets. But I think manually annotating the 3D keypoints is quite difficult, especially under sub-mm accuracy. Is there any technique to ease this problem?*
>
> ### Annotation of distal limbs:
> We agree that including annotations of points on distal limbs would be valuable for the field. We have several strategies for collecting such data that we discuss in the “Remarks to all Reviewers.”

---

### Official Review · Reviewer_HKjG · 2021-07-04
**Review on Submission 172**

**Rating:** 7
**Confidence:** 3
**Clarity:** The paper is overall well-written and…

**Strengths:**

a. It is valuable as the first multi-animal 3D pose benchmark to advance the progress in this area.

b. The large scale and diversity of the collected data could advance progress in this area.

c. The semi-automated labeling of behavior is interesting and meaningful for analysis on animals, and further fine-grained labeling of behavior is promised which would be very helpful.

**Weaknesses:**

a. The lack of hand, foot and limb annotation limits full understanding of animal pose as the authors stated in Section 4. I'm looking forward to the completion of these annotations.

b. It seems to me that the markers totally occlude the rats' head. Would it harm the generalization when the rats wear no markers?

Updated after rebuttal:
The authors response addressed my main concerns.

**Additional Feedback:**

It would be better if some of the failure cases (frames discarded for severe occlusion) could be illustrated.

Updated after rebuttal:
The authors promised to provide these samples.

**Correctness:**

I believe the claims are correct. The dataset is constructed soundly. The experiment is well-designed and appropriate.

**Documentation:**

The data collection and organization, availability and maintenance, and ethical and responsible use are sufficiently provided. The authors have provided details for reproduction. The URL for the metadata of the dataset is accessible.

**Ethics:**

There are no ethical concerns that warrant further discussion or review.

**Relation To Prior Work:**

Sufficient discussion on prior works is given.

**Summary And Contributions:**

To address the lack of standardized benchmarks for multi-animal 3D pose estimation, the author collects the first large-scale multi-animal 3D pose benchmark with multiple viewpoints and distinct rat pairs. Ground truth 3D pose annotation is provided via motion capture. Also, behavioral labels are given in a semi-automated way. A strong baseline for multi-animal 3D pose estimation without motion capture is given via an adapted version of DANNCE.

---

> ### Author Response · Authors · 2021-07-13
> **Author Response**
>
>
> We thank the reviewer for their critical commentary and praise. We have addressed each of the comments below and will incorporate them into future versions of the text.
>
> > *The lack of hand, foot and limb annotation limits full understanding of animal pose as the authors stated in Section 4. I'm looking forward to the completion of these annotations.*
>
> ### Additional points on hands and feet:
> We agree with the reviewer that a multi-animal dataset with additional points on the hands and feet would be a valuable asset to the field. We have plans to collect such a dataset in the future, but we will also automatically label these distal points using interleaved training on the PAIRS-R24 dataset and the Rat7M dataset, which was collected in (Dunn et al. 2021). We comment on this in more detail in the “Remarks to all Reviewers.”
>
> > *It seems to me that the markers totally occlude the rats' head. Would it harm the generalization when the rats wear no markers?*
>
> ### Generalization:
> We think that this is a good point, although we don’t feel the headcap limits the use of the dataset as a benchmark for pose estimation algorithms. Furthermore, while the headcap may limit the immediate generalization of the network to non-headcap recordings, we have shown that with a fairly low number of additional examples (50-100), the network can be fine-tuned to work in new domains, and on animals without headcaps. We demonstrated this in previous work (Dunn et al. 2021), where we trained DANNCE using a large dataset of rats with headcaps bearing 20 markers (Rat7M), and fine-tuned it to work in mice, marmosets, birds, and rats not bearing markers.
>
> > *It would be better if some of the failure cases (frames discarded for severe occlusion) could be illustrated.*
>
> ### Failure cases:
> In the revised manuscript we will give more examples of cases where frames are not included due to occlusions, and show DANNCE predictions on these frames. Please also see the “Inter-animal occlusion” response to Reviewer 1.

---

### Official Review · Reviewer_5CY4 · 2021-07-06
**The PAIR-R24M Dataset for Multi-animal 3D Pose Estimation**

**Rating:** 7
**Confidence:** 3

**Strengths:**

In stark contrast with previous multi-animal pose estimation datasets, the main strength of this dataset is that it provides three-dimensional annotations. Additionally, the number of annotated videos and their dimension are bigger than similar datasets. Finally, the authors performed a strong benchmark by adapting DANNCE[1], a model for individual animal pose estimation to the multi-animal setting.

[1] Timothy W Dunn, Jesse D Marshall, Kyle S Severson, Diego E Aldarondo, David GC Hildebrand, Selmaan N Chettih, William LWang, Amanda J Gellis, David E Carlson, Dmitriy Aronov, et al. Geometric deep learning enables 3d kinematic profiling across species and environments.

**Weaknesses:**

As the authors already stated, one of the main concerns consists in the number of markers used to keep track of the pose. Since they do not cover all the important points on the body of the animal, there might be problems in identifying the corresponding behaviour. Another important issue, that is a general challenge for this type of dataset, is the occlusion that inevitably happens while the animals interact with each other. Furthermore, sharing more insights about the way the labels of the behaviour were selected would better position this work.  Were the categories defined by the authors or are they a standard in the field? Also, a benchmark on the behaviour classification would have been nice.

**Additional Feedback:**

-

**Clarity:**

The paper is well organized and the writing is clear. However, I would suggest changing the name of Section 3, as in my opinion "Results" sounds too general and not really correlated to creating a dataset.

**Correctness:**

Regarding the pose estimation part, the dataset is constructed in an objective way, using body landmarks to collect the corresponding information. With respect to the specific and interaction behavioural categories, I consider more details should have been provided.

**Documentation:**

Yes, the authors provide necessary details about data collection and annotation, together with the URL giving access to the dataset.

**Ethics:**

It is stated that "The care and experimental manipulation of all animals were reviewed and approved by the Harvard University Faculty of Arts and Sciences Institutional Animal Care and Use Committee". Even if it justifies the ethics, providing the report of the mentioned committee would be appreciated.

**Relation To Prior Work:**

The differences with previous works are clearly discussed in the related work section. It presents 2D and 3D pose estimation datasets, both in terms of single and multi-animal settings, together with their limitations. The contributions and the importance of the current dataset are also mentioned.

**Summary And Contributions:**

This paper introduces a new large scale dataset for multi-animal 3D pose estimation. It consists of approximately 22 million frames capturing interactions between 18 pairs of laboratory rats. The videos are captured from 30 viewpoints, using 6 cameras placed at 5 different locations and annotations are collected using body markers that provide information about the heads, spines, trunks, hips, and shoulders of the animals. The videos are also annotated for the action recognition task, with 11 behavioural categories and 3 inter-animal interaction categories. Furthermore, a strong baseline for 3D pose estimation is provided.

---

### Author Response · Authors · 2021-07-13
**Remarks to all reviewers**

We thank all reviewers for their comments, which we were glad to see were very positive overall. The reviewers provided constructive feedback and raised important points, which we address in full below.

All reviewers were excited by the prospect of future work introducing an expanded marker set that includes points on the distal limbs.

- Reviewer 1 mentioned that these additional markers will be important for more precisely defining behavioral classes, “ *As the authors already stated, one of the main concerns consists in the number of markers used to keep track of the pose. Since they do not cover all the important points on the body of the animal, there might be problems in identifying the corresponding behaviour.* ”

- Reviewer 2 stated, “ *The lack of hand, foot and limb annotation limits full understanding of animal pose as the authors stated in Section 4. I'm looking forward to the completion of these annotations.* ”

- Reviewer 3 stated, “ *Line 261, the authors mention human annotations of distal limbs can be added in future datasets. But I think manually annotating the 3D keypoints is quite difficult, especially under sub-mm accuracy. Is there any technique to ease this problem?* ”

We agree that these additional markers will be an important future contribution to the dataset. We see 3 potential strategies for delivering these data,

1. Adding additional motion capture markers on the distal limbs, and collecting new data.
2. Crowd-sourcing manual annotations for these additional landmarks. This approach is used successfully by several popular human 3D pose datasets (e.g. the Campus and Shelf datasets in Belgiannis et al. CVPR 2014).
3. Using a 3D pose estimation algorithm to provide these additional landmarks. This approach is used successfully by several popular human 3D pose datasets (e.g. for the CMU panopticon dataset described in Joo et al. 2019, IEEE PAMI and in MPI-INF-3DHP dataset in Mehta et al. 3DV 2017) and by social 2D pose datasets (e.g. Sun et al. arXiv:2104.02710).

Strategy (1) will require additional data collection and the development of new approaches to assign limb landmarks to the correct subjects during motion capture post-processing. Thus, we believe this is best suited for a follow-up publication.

We note that for (2), while this is unlikely to achieve sub-mm accuracy (without averaging over many human labelers), c.f. Reviewer 3, it is possible to achieve mm-scale accuracy smaller than the size of the hands, feet, elbows, and knees. These are the default limb landmarks that we have tracked in the past (Dunn et al. Nature Methods 2021, Marshall et al. Neuron 2021) and are sufficient to robustly identify large and diverse sets of behavioral classes.

Although we interpret the reviewer comments as being hopeful for these additional markers in future work, strategy (3) is feasible now, and we will add inferred limb 3D positions to the dataset. This will require training a version of DANNCE that outputs not just the 10 landmarks reported in the current benchmark but also the additional limb landmarks. To do this, we will shuffle our previous Rat 7M single-animal dataset (from Dunn et al. 2021 Nature Methods), which includes 20 landmarks comprehensive of the limbs, into our PAIRS dataset and train a 20-landmark DANNCE network. This will allow DANNCE to retain its predictive accuracy on limb landmarks while also learning to overcome occlusions specific to the multi-animal setting. If necessary, we can also introduce a relatively small set of manually-labeled 20-landmark multi-animal data to augment training, which our qualitative results in markerless animals suggest is sufficient for learning to infer a full set of landmarks. Our previous work suggests that our limb predictions will achieve 90-95% accuracy (Dunn et al. Nature Methods 2021), and we will validate a pose-balanced subset of these inferred limb landmarks against a committee of human labelers.

This strategy of using an algorithm to provide labels has been used broadly by the pose dataset community and has sustained effective method development for pose estimation and action recognition in humans (e.g. for the CMU panopticon dataset described in Joo et al. 2019, IEEE PAMI) and animals (e.g. Sun et al. arXiv:2104.02710).

---

### Decision · Program_Chairs · 2021-07-26

**Decision:**

Accept

**Comment:**

Reviewers have an overall positive opinion regarding the paper contribution. Overall the provided data set is large, unique in 3D animal poses, and provides useful complementary annotations. This can be a valuable resource for the community.